# The Role of Electrochemical and Spectroelectrochemical Techniques in the Preparation and Characterization of Conjugated Polymers: From Polyaniline to Modern Organic Semiconductors

**DOI:** 10.3390/polym14194173

**Published:** 2022-10-05

**Authors:** Przemyslaw Ledwon, Mieczyslaw Lapkowski

**Affiliations:** 1Department of Physical Chemistry and Technology of Polymers, Faculty of Chemistry, Silesian University of Technology, Strzody 9, 44-100 Gliwice, Poland; 2Centre for Organic and Nanohybrid Electronics, Silesian University of Technology, Konarskiego 22B, 44-100 Gliwice, Poland; 3Centre of Polymer and Carbon Materials, Polish Academy of Sciences, M. Curie-Sklodowskiej 34, 41-819 Zabrze, Poland

**Keywords:** electrochemical polymerization, spectroelectrochemistry, conjugated polymers, polymerization mechanism, polyaniline, organic semiconductors, electrochromism

## Abstract

This review article presents different electrochemical and spectroelectrochemical techniques used to investigate conjugated polymers. The development of this research area is presented from an over 40-year perspective—the period of research carried out by Professor Mieczyslaw Lapkowski. Initial research involved polymers derived from simple aromatic compounds, such as polyaniline. Since then, scientific advances in the field of conductive polymers have led to the development of so-called organic electronics. Electrochemical and spectroelectrochemical methods have a great influence in the development of organic semiconductors. Their potential for explaining many phenomena is discussed and the most relevant examples are provided.

## 1. Introduction

The development of conjugated polymers began with the discovery of a dramatic increase in the conductivity of doped polyacetylene in 1977 [1]. This breakthrough discovery was noticed at the Silesian University of Technology at the Department of Physical Chemistry and Technology of Polymers, and, soon, the first scientific work on these polymers began here. Professor Łapkowski was at that time and remains a pioneer in this field, not only at the Silesian Technical University but also in Poland. During over 40 years of research on conductive polymers, numerous research methods have been developed, and many new polymer structures have been synthesized.

Electrochemical methods have, for many years, been widely used for preparing conjugated polymers. Electrochemical oxidation is a popular method of obtaining conjugated polymers. Electrochemical oxidation was originally used to polymerize simple heterocyclic compounds and aromatic hydrocarbons [2]. Especially often, polythiophenes, polypyrroles and polyaniline were obtained via electrochemical oxidation of the corresponding monomers.

Subsequent research began to involve chemically modified monomers. The research included various modifications, typically in the form of functionalization with different substituents, e.g., alkyl and alkoxy groups, which were introduced to improve the solubility of conjugated polymers in common organic solvents.

The following years brought about the development of conjugated co-polymers. The electronic and optical properties of the obtained co-polymers can be modified to a very wide extent by using different monomers. Electrochemical deposition and typical chemical methods can be used for this purpose. It should be noted, however, that electrochemical co-polymerization should be approached highly critically. Although copolymers can indeed be produced via the polymerization of a co-monomer mixture, it is more common to instead obtain homopolymer blends or altogether different products from the intended copolymers [3]. In this aspect, extensive structural investigation of the obtained “copolymer” is essential and many tools are available for this purpose [4,5].

Polymers with a donor–acceptor structure were found to be a particularly interesting group of compounds. The term “donor–acceptor structure” refers to a conjugated system, in which electron-poor (electron acceptor) units are combined with electron-rich (electron donor) units. These types of compounds have caused the rapid development of organic electronics and photovoltaics. This is due to the large possibilities of controlling energy levels, as well as p- and n-type conductivity [6].

Over the course of developing conjugated polymers as an individual research field, a need for more in-depth investigation of the tested materials and occurring phenomena arose. Electrochemical measurements were coupled with different spectroscopic techniques, so as to develop spectroelectrochemical investigation methods and protocols. The spectroelectrochemical methods commonly used to investigate conjugated polymers often combine electrochemical methods with ultraviolet–visible–near-infrared (UV–Vis–NIR) spectroscopy, electron paramagnetic resonance (EPR) spectroscopy, Raman spectroscopy, etc. Electrochemical and spectroelectrochemical research allowed the determination of the future directions of practical applications of conductive polymers. In this context, it should be mentioned that the initial predictions about conjugated polymers speculated on their potential application as organic conductors. Numerous electrochemical studies, however, have shown the low stability of organic materials in their highly doped states, largely compromising such an application. Simultaneously, however, conjugated polymer materials were found to exhibit very good semiconducting properties. This has opened many other possible routes for the development of organic semiconductor materials. A new, huge branch of science called organic electronics has been created. Today, it covers a broad array of important research areas, such as organic materials for photovoltaics, organic light-emitting diodes, touch panels, transistors, electrochromic devices and sensors.

The development of π-conjugated polymers and the use of electrochemical and spectroelectrochemical techniques for the investigation of such materials is presented from the perspective of the 40+-year research activity period of Professor Mieczyslaw Lapkowski. Significant achievements included, in particular, the study of the mechanisms of polyaniline formation and the doping mechanisms of many polymers. Years of research have led to many organic semiconductors being tested in applications such as organic light-emitting diodes (OLEDs) [7,8] sensors [9,10,11] and solar cells. Electrochemical and spectroelectrochemical techniques used for investigating the properties of conjugated polymers are briefly presented in the manuscript. The description of these techniques is accompanied by numerous publications showing the practical aspects of their application and exemplifying research problems that were resolved using these methods.

## 2. Electrochemical and Spectroelectrochemical Techniques Used in the Preparation and Characterization of Conjugated Polymers

### 2.1. Electrochemical Polymerization

Electrochemical techniques played an extremely important role in the synthesis of the first conductive polymers. Cyclic voltammetry, in particular, was used extremely commonly in the initial research to effect direct polymer synthesis on the surface of a working electrode [12]. Such important conjugated polymers as polythiophene, polypyrrole and polyaniline were obtained electrochemically.

Electrooxidation by cyclic voltammetry allows the conjugated polymer to be directly deposited at a working electrode. Subsequent oxidation cycles induce a repetition of this process, leading to the deposition of subsequent polymer layers. This process leads to the formation of progressively thicker layers, therefore manifesting as a current that increases with each cycle, as in the case of the electrochemical deposition of polyaniline (Figure 1).

In this case, the polymerization process can be carefully controlled, via controlling the applied potential, current flowing through the electrode and polymerization charge [14,15]. All these factors are often important for the structure of the polymer and the quality of the obtained polymer layers [16].

Electrochemical polymerization is a cost-effective and easy-to-use method for preparing conjugated polymers [17,18]. Currently, it is particularly eagerly used to obtain electrochromic films [19], sensors [20] and energy storage materials [21,22]. Attempts are made to obtain polymers for other modern applications, such as OLEDs [23] and dye-sensitized solar cells (DSSCs) [24]. Despite its many advantages, electrochemical polymerization has limitations. The insolubility or limited solubility of the obtained polymers is one of the most important factors [17]. In the case of some aromatic compounds, electrochemical oxidation leads to side reactions other than polymerization, which prevents the formation of well-defined polymers on the electrode surface. Preparing complex polymers and establishing control over their structure is also problematic, because copolymerization can lead to structures different from those intended [12].

### 2.2. Electrochemical Studies of the Polymerization Mechanism by the Example of Polyaniline

The mechanism of electrochemical deposition of conjugated polymers was of interest to many research groups [25,26,27]. Based on these publications, differing mechanisms were proposed, depending on the heterocyclic compound and employed experimental conditions. This is well exemplified by the comparison of the mechanisms postulated for the polymerization of simple heterocyclic compounds, such as pyrrole or thiophene, with mechanisms proposed for the polymerization of multi-cycle compounds, such as carbazole [12,28,29]. In the case of simple heterocyclic compounds and some more complex compounds, two electrons per monomer molecule are required to cause polymerization [25]. Oxidation leads to the formation of radical cations, which undergo recombination, leading to the formation of polymers in subsequent steps [30].

Upon achieving such molecular weight that it is no longer soluble in the supporting electrolyte/solvent medium, the incipient polymer precipitates onto the surface of the working electrode, a feature that can be utilized to fine-tune the chemical structure and properties of conjugated polymers produced via electrochemical polymerization [31,32]. In some cases, radical cations are highly stable and further oxidation is necessary—for example, in the case of macromonomers with extensive systems of pi bonds [19].

The mechanism of the electrochemical polymerization of aniline differs from the mechanism postulated for five-membered heterocyclic systems (e.g., pyrrole, thiophene). Research conducted in the 1980s was focused on explaining this mechanism.

The preparation of polyaniline has shown that one of the most important factors is the maximum oxidation potential applied to the monomer [13,33]. A typical course of the voltammetric polymerization of aniline in sulfuric acid on a metal or optically transparent SnO_2_ electrode (Figure 1) has been observed by many researchers dealing with the electrochemistry of polyanilines. The presence of two small redox systems designated as II—II’ should be noted, as the explanation of their presence was controversial for many years. Some scientists believed that the presence of these peaks was related to the presence of the quinone–hydroquinone redox system, formed as a result of the hydrolysis of oxidized polyaniline. The second group believed that this redox system was caused by errors in the polyaniline structure, associated with the formation of ortho bonds or the formation of phenazine structures. Figure 2 shows our synthesis of polyaniline in tetrafluoroboric acid. In this case, by limiting the potential to the beginning of the monomer oxidation potential, it was possible to obtain polyaniline devoid of the II—II’ redox systems, i.e., structural defects [34]. It can therefore be said that, in this case, linear (lacking ortho couplings) polyaniline is obtained.

It is generally accepted that the first step in the oxidation of aniline is the formation of a radical cation, which is independent of the pH of the synthesis medium and is resonance-stabilized by the canonical forms presented in Figure 3.

It was assumed that the aniline polymerization reaction proceeds according to the classical ECE mechanism (electron transfer, chemical reaction, electron transfer), where, in the first stage, two radical cations in the forms I and III react with each other to give the dimer—p-aminodiphenylamine (PADPA)—after deprotonation. The dimer is oxidized at a less positive potential than aniline, i.e., it forms PADPA^+^, which, in the next steps, reacts with the aniline radical or PADPA^+^. This is shown in Figure 4.

The polymerization process proceeds according to this scheme only when the reaction is carried out at potentials close to the beginning of the monomer oxidation peak, as shown in Figure 2. When the potential is higher, more aniline radical cations are produced and the probability of radical cation I reacting with radical cations II and IV increases, causing “ortho errors”. Using UV–Vis spectroelectrochemistry, it was also shown that increasing the oxidation potential to 1 V causes the formation of another reactive product of aniline oxidation—the nitrenium cation—which can react with the monomer, the radical cations and even with the polymer to give phenazine structures [13,35].

A linear polyaniline structure, i.e., having two redox systems, can be converted into a polymer containing phenazine structures by increasing the end-scan potential well above that of the second redox system, i.e., above 1.0 V. Figure 5 shows cyclic voltammograms of polyaniline in anhydrous NH_4_F:2.3 HF liquid salt solution.

As can be seen in Figure 5, peaks III and IV can be correlated with the peaks marked with II in Figure 1. It follows that phenazine structures are formed both during the polymerization and during the oxidation of the linear polymer at high potential, which can be described by the reaction scheme presented in Figure 6 [36].

The above demonstrates that polyaniline evolution moves towards a two-dimensional polymer with phenazine rings, which can be formed by a cross-linking reaction (Figure 6) or by the insertion of nitrenium cations of aniline [35,36].

### 2.3. Spectroelectrochemical Methods

Spectroelectrochemical measurement combines the simultaneous use of electrochemical and spectral techniques (Figure 7). Using spectroscopic methods, it is possible to study changes that occur in the measured sample as a result of changing the electrochemical potential. The most frequently used technique is UV–Vis–NIR spectroelectrochemistry. However, the number of spectral techniques that can be used for in situ measurements in combination with electrochemical measurements is much greater. Examples include transmission spectroscopic ellipsometry [37], nuclear magnetic resonance [38], EPR and infrared and Raman spectroscopy [39]. The coupling of electrochemical and spectroscopic techniques provides complementary information. For example, such measurements can help in studying the processes occurring during the oxidation or reduction of conductive polymers, such as doping and dedoping [37,40], degradation [41], electrochromism [42], intermolecular interactions [38,43] and mechanisms of synthesis [44].

The typical spectroelectrochemical measurement setups are shown in Figure 8. Typical electrochemical setups include working, counter and reference electrodes. In the case of spectroelectrochemical measurements, the set of electrodes is essentially the same [45]. However, the electrodes must be compatible with the measuring cells and experimental design. As an example, during UV–Vis–NIR spectroelectrochemical measurement in transmittance mode, optically transparent working electrodes, e.g., in the form of transparent electrodes, such as indium-tin oxide (ITO) electrodes, or in the form of a mesh with fine holes, must be used. In turn, in the case of EPR spectroelectrochemistry measurements, it is favorable for the working electrode to be elongated, allowing the amount of solvent present within the EPR spectrometer resonator to be minimized, so as to avoid compromising the sensitivity of the instrument.

### 2.4. Time-Resolved Spectroelectrochemical Methods

An important point of utility in the use of spectroelectrochemical methods is that most spectroscopic methods can operate in time-resolved mode. In this mode, rapid acquisition of successive spectra during the course of electrochemical processes needs to be achieved. This allows a variety of electrochemical and electrochemically induced processes to be closely followed spectrally. Such spectral monitoring can be utilized for various purposes, such as for observing and identifying reaction intermediates and interactions between polymers [38].

In relation to conducting polymers, monitoring the electropolymerization and doping/dedoping processes is a particularly valuable use of time-resolved spectroelectrochemical methods [46]. In the aspect of electropolymerization, the key advantage is that the thickness and electrochromic properties of the growing films can be initially assessed, particularly when utilizing potentiodynamic electrochemical methods. For example, fluorescence spectroelectrochemical measurements were used to study the electropolymerization process, revealing competing processes taking place in the measurement system [47].

In the case of a polymer’s doping/dedoping processes, the use of time-resolved spectroelectrochemical methods allows investigation of the kinetics of doping/dedoping processes, as opposed to non-time-resolved spectroelectrochemical methods. Time-resolved UV–Vis–NIR spectroelectrochemical measurements are commonly used to analyze electrochromic materials. The electrochromic switching time and the long-term switching stability are important parameters that can be estimated by transmittance changes during electrochemical switching recorded in time-resolved mode [19,48].

### 2.5. EPR, UV–Vis–NIR and Raman Spectroelectrochemical Measurement of Conjugated Polymers

Electrochemical deposition allowed the direct electrochemical characterization of the obtained polymer films. As a result, it was possible to understand the phenomena occurring during the oxidation of polymers. It was especially important to explain the phenomenon of an increase by several orders of magnitude in the conductivity of the layers of conductive polymers occurring during oxidation. This phenomenon was called doping, by analogy to the physics of inorganic semiconductors. Numerous studies aimed at understanding the doping phenomena occurring in conjugated polymers have significantly contributed to the development of organic semiconductors [38,49]. The combination of electrochemical methods and spectroscopic techniques is especially important in these studies.

A particularly frequently used technique is UV–Vis–NIR spectroelectrochemistry. This is due to the wide availability of UV–Vis/UV–Vis–NIR spectrometers. An example of UV–Vis–NIR spectra recorded during a UV–Vis–NIR spectroelectrochemical experiment is shown in Figure 9 [19]. Spectra recorded at different potentials, related to different redox states, can indicate which moiety of the polymer is reacting. It can be stated whether the whole molecule is reacted or only a part of it.

EPR spectroelectrochemistry is another prominently used method, which can in turn detect spin-bearing species during electrochemical oxidation or reduction (Figure 10). An important highlight of EPR spectroelectrochemical methods is the ability to use the EPR data to analyze which atoms constituting the conjugated system the charge carriers interact with and to estimate the degree of their delocalization over the conjugated bond system [50,51,52,53]. EPR spectroelectrochemical data can be used for the study of the redox processes of compounds with ambipolar characteristics; for example, it can confirm the differences in the locations of cation radicals and anion radicals generated, respectively, during oxidation and reduction processes [54]. Charge-trapping effects and π–π interactions can also be estimated based on EPR spectroelectrochemical data [55].

Degradation processes can be also observed using EPR spectroelectrochemical measurement. Over-oxidation of conjugated polymers is an important issue that leads to limited stability. EPR spectra can help to elucidate this problem. The degradation of conjugated polymer poly(3,4-ethylenedioxythiophene) PEDOT was studied by EPR during electrochemical oxidation [57]. Such phenomena as cross-linking, decreases in the conjugation length and spins trapped in isolated packets can be tracked.

### 2.6. Spectroelectrochemical Conductometry

Modern approaches to in situ conductometry stem from two- [58] and four-point [59] probe methods, transitioning through the use of multiple band electrodes and currently relying primarily on interdigitated array electrodes (IDEs) [60]. IDEs are typically employed as a double-working-electrode system, onto which the conjugated polymer layer is deposited. The measurement involves offsetting the electrical potentials applied to the two working electrodes (arrays) and observing the currents flowing through the two working electrodes [61], with the conductivity being calculated via a simple mathematical operation, taking into account the geometry and dimensions of the IDE [62]. IDEs can be deposited on different substrates, including transparent substrates, which enables UV–Vis spectroelectrochemical conductometry measurements.

The issue of the doping/dedoping of conjugated polymer layers is inevitably tied to changes in their conductivity. Practically, the use of electrochemical conductometry allows some initial insight into the changes taking place in conjugated polymer films during doping. Such insights, while worthwhile, should be considered critically and followed up by more thorough investigation. As an example, an increasing doping level was accompanied by a decline in conductivity for a fairly standard poly(3-hexylthiophene)-based polymer, which would intuitively, and based on the literature, be explained by the degradation of the polymer. However, based on extensive spectroelectrochemical investigation, a hypothesis that the transition from polarons to bipolarons as the primary charge carrier was accompanied by a loss of charge carrier mobility was proposed and eventually confirmed [63].

Due to the greater impact of the presence of charge carriers on the macroscopic conductance of conjugated polymer films than on their spectral features, conductometry has also been used to identify the presence of minor doping state hystereses in materials that were otherwise considered as lacking such hysteresis, based on spectroelectrochemical data [63,64].

An important consideration for following the doping/dedoping process is that of sensitivity. Although typical spectroscopic methods (e.g., UV–Vis–NIR, EPR) allow the detection of the absorption bands associated with the presence of charge carriers, their sensitivity is severely limited, both due to instrumental factors and the limited populations of such charge carriers, particularly at the initial stages of doping, being unable to give a magnitude of absorption signals that allows differentiation between them and random noise [65].

Conversely, even such limited charge carrier populations can result in a noticeable change in the macroscopic conductivity of a conjugated polymer layer (Figure 11). As such, the already valuable information obtained from conductometric methods can be further utilized to follow the doping/dedoping process far more closely than using typical spectroelectrochemical methods.

Despite numerous methods of monitoring the conductivity of a polymer layer, both ex situ and in situ, methods combining conductometry and other measurement techniques were remarkably rare, and primarily focused on utilizing impedance measurements [66].

### 2.7. Electrochromic Characterization

Electrochromism is a phenomenon in which changes between different colored states and/or colorless states occur under external electrical bias [67]. Moreover, electrochromism is not restricted to the visible part (Vis) of the electromagnetic range but also refers to changes in transmittance in the near-infrared (NIR) and ultraviolet (UV) range [68]. Among the materials exhibiting these properties, inorganic ones, such as Prussian blue [69,70] and tungsten oxide [71], can be mentioned. However, compared to these traditional materials, conductive polymers constitute a larger group of compounds [72]. The two main groups of polymers for this application are donor–acceptor polymers and poli(3,4-etyleno-1,4-dioksytiofenand its derivatives [73]. The changes in the transmittance of conjugated polymers in the UV, Vis and NIR range are related to the changes in structure and orbitals of these materials under the influence of doping. Current trends in research on electrochromic polymers focus on materials that can use more than two color states [19]. Typically, different doping states may result in different colors, leading to multielectrochromic layers.

The combination of UV–Vis–NIR spectroscopy and electrochemistry is crucial for the characterization of electrochromic materials. As mentioned in the previous section, UV–Vis–NIR spectroscopy enables the registration of UV–Vis–NIR. The doping level can be precisely controlled using different electrochemical techniques. For each oxidation or reduction step, it is possible to determine the absorbance/transmittance of the studied material in a broad range, as shown in Figure 10. Another important aspect is the determination of color parameters, such as the CIE 1931 XYZ color space created by the International Commission on Illumination.

In the case of materials for electrochromic applications, it is important that the processes of doping and dedoping are reversible and electrochemically controlled. Kinetic research is very useful in this aspect. Kinetic studies can be carried out by registering the transmittance for a specific wavelength λ, and the charge flow during multiple doping and dedoping steps of the electrochromic layer is recorded.

## 3. Conclusions

More than 40 years have passed since the discovery of the conductivity phenomenon in doped conjugated polymers. Since then, there has been intensive development in this class of organic materials. The combination of various electrochemical and spectral techniques makes a significant contribution to the development of this class of materials. Thanks to them, it is possible to learn about the phenomena taking place in polymers, such as the processes of doping, degradation, electrochromism and reaction mechanisms. Examples are provided in the manuscript. In particular, the article summarizes research articles focused on the elucidation of the mechanism of electrochemical generation and doping of polyaniline. The examples of the use of different electrochemical and spectroelectrochemical techniques shown in the manuscript indicate the further wide possibility of their application. UV–Vis–NIR spectroelectrochemistry, in particular, is broadly used.

In the coming years, electrochemical polymerization is expected to achieve wider practical application as a cost-effective and easy-to-use preparation method. This is especially so as interest in conductive polymers continues to increase. These materials are the foundation of modern organic semiconductors. In recent years, we have seen a continuous improvement in the performance of conductive polymers in various applications, such as different types of photovoltaic cells, sensors and light- and NIR-emitting diodes, and we expect further groundbreaking development to take place.

## Figures and Tables

**Figure 1 polymers-14-04173-f001:**
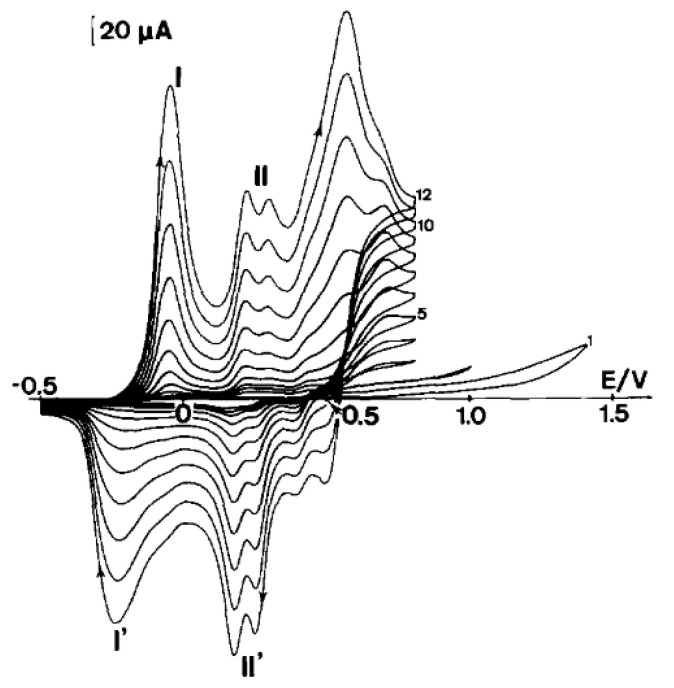
Growth of polyaniline film from a 0.1 M aniline solution in 1 M sulfuric acid achieved via cyclic voltammetry. Sweep rate was 50 mV/s; electrode area was 1 cm^2^. Reprinted from [13], Copyright 1990, with permission from Elsevier.

**Figure 2 polymers-14-04173-f002:**
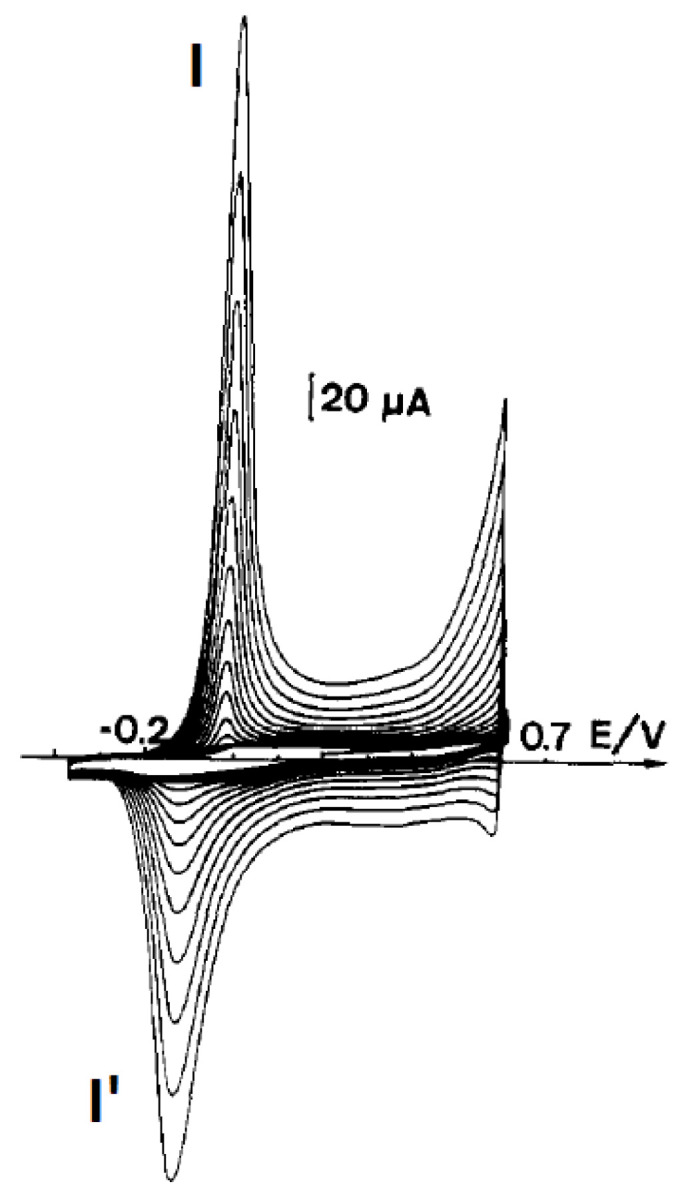
Growth of polyaniline film from a 0.5 M aniline in 35% HBF_4_ during voltammetric cycling at 50 mV s^−1^. Reprinted from [34], Copyright 1990, with permission from Elsevier.

**Figure 3 polymers-14-04173-f003:**
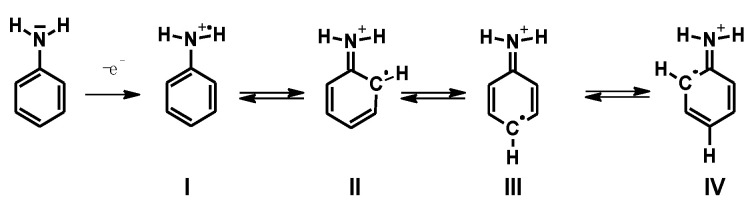
Canonical forms of aniline radical cation.

**Figure 4 polymers-14-04173-f004:**
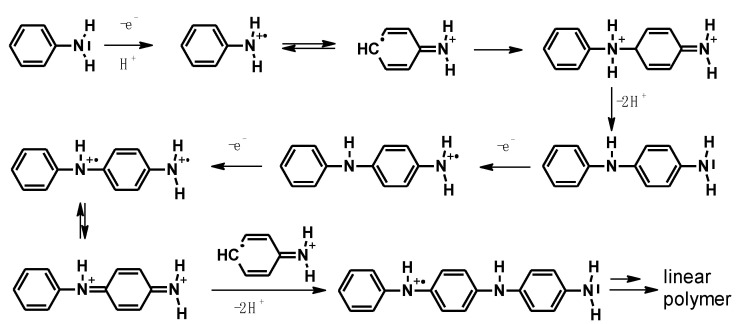
Reaction mechanism for radical cation coupling of aniline, leading to the forms of a linear polymer.

**Figure 5 polymers-14-04173-f005:**
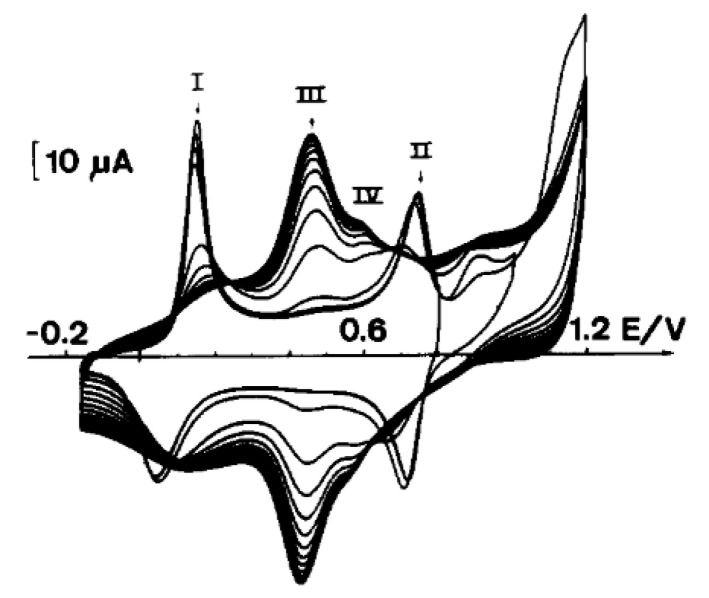
Modifications in the cyclic voltammetric curves at 100 mV/s on a polyaniline film during cycling, from −0.2 V to 1 and 1.2 V in NH_4_F:2.3 HF. Reference electrode: Cu/CuF. Reprinted from [36], Copyright 1988, with permission from Elsevier.

**Figure 6 polymers-14-04173-f006:**
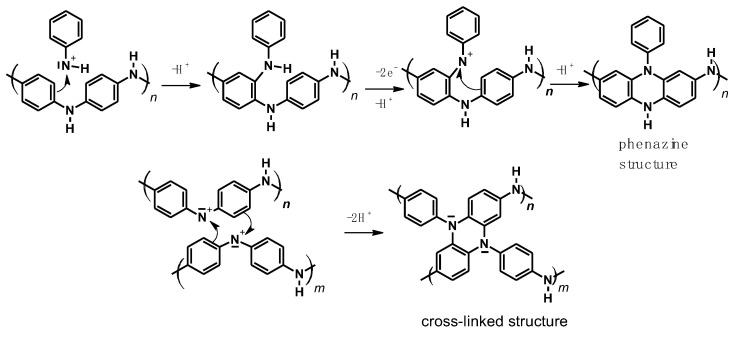
The reactions of polyaniline during oxidation leading to the formation of phenazine structures and cross-linked structures [36].

**Figure 7 polymers-14-04173-f007:**
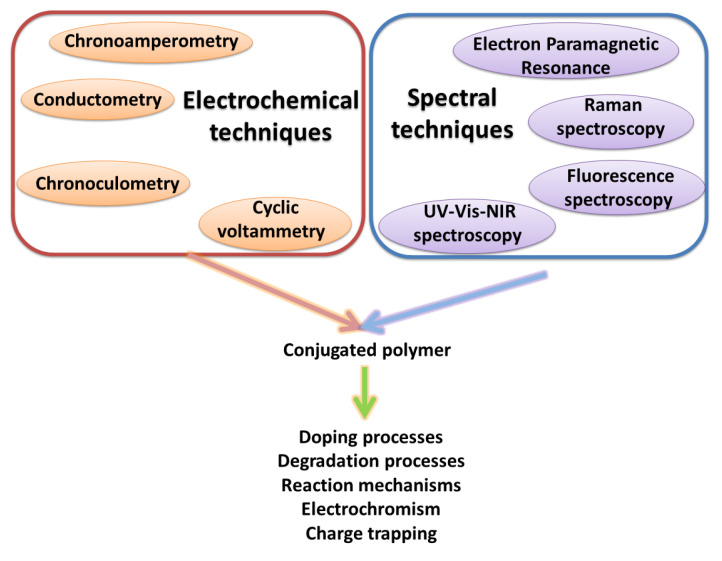
Graphical presentation of different combinations of electrochemical and spectral methods used in spectroelectrochemistry.

**Figure 8 polymers-14-04173-f008:**
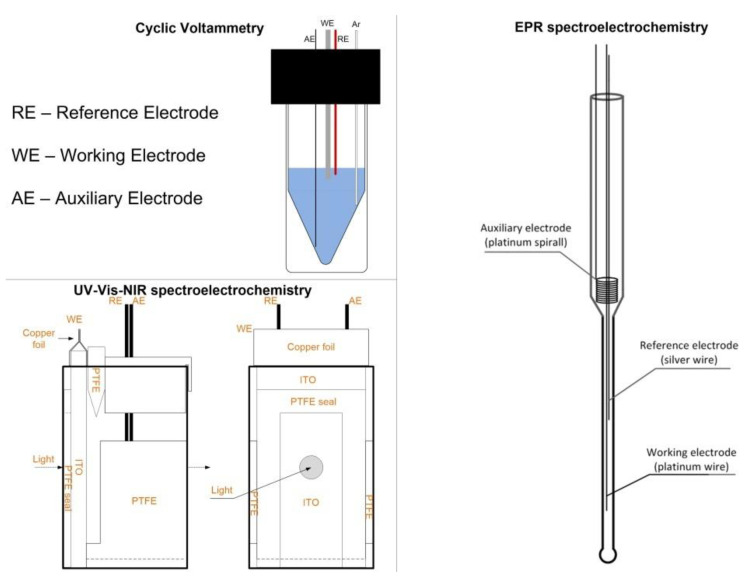
Electrochemical and spectroelectrochemical cells used for measurements. The figure presents the scheme setup of electrochemical/spectroelectrochemical cells using cyclic voltammetry, ultraviolet–visible and near-infrared (UV–Vis–NIR) and electron paramagnetic resonance (EPR) spectroelectrochemical measurements. This is adapted from Pluczyk, S., Vasylieva, M., Data, P. Using Cyclic Voltammetry, UV-Vis-NIR, and EPR Spectroelectrochemistry to Analyze Organic Compounds. J. Vis. Exp. (140), e56656, doi:10.3791/56656(2018) [45].

**Figure 9 polymers-14-04173-f009:**
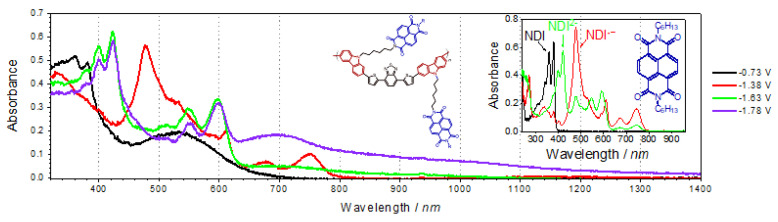
UV–Vis–NIR spectra recorded at different potentials. Reprinted with permission from [19]. Copyright 2019 American Chemical Society.

**Figure 10 polymers-14-04173-f010:**
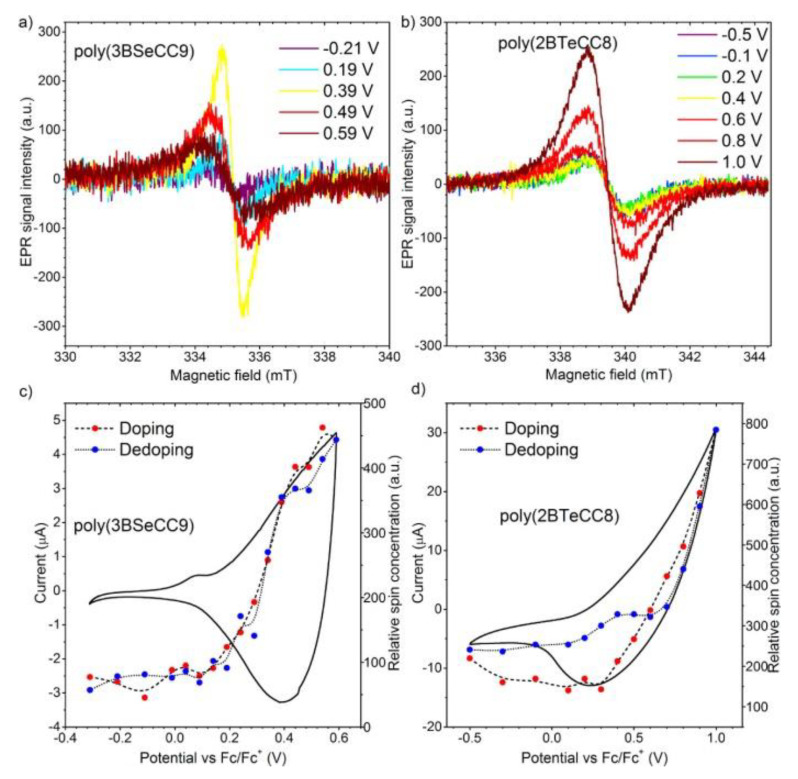
Selected EPR spectra measured during doping of a thin film of (**a**) poly(3BSeCC9) and (**b**) poly(2BTeCC8). Relative spin concentration during doping (red circles) and dedoping (blue circles) of a thin film of (**c**) poly(3BSeCC9) and (**d**) poly(2BTeCC8). Recorded in 0.1 M Bu_4_NBF_4_/DCM supporting electrolyte solution. Potentials recorded relative to ferrocene/ferrocenium (Fc/Fc^+^) redox couple. Reprinted with permission from [56]. Copyright 2017 American Chemical Society.

**Figure 11 polymers-14-04173-f011:**
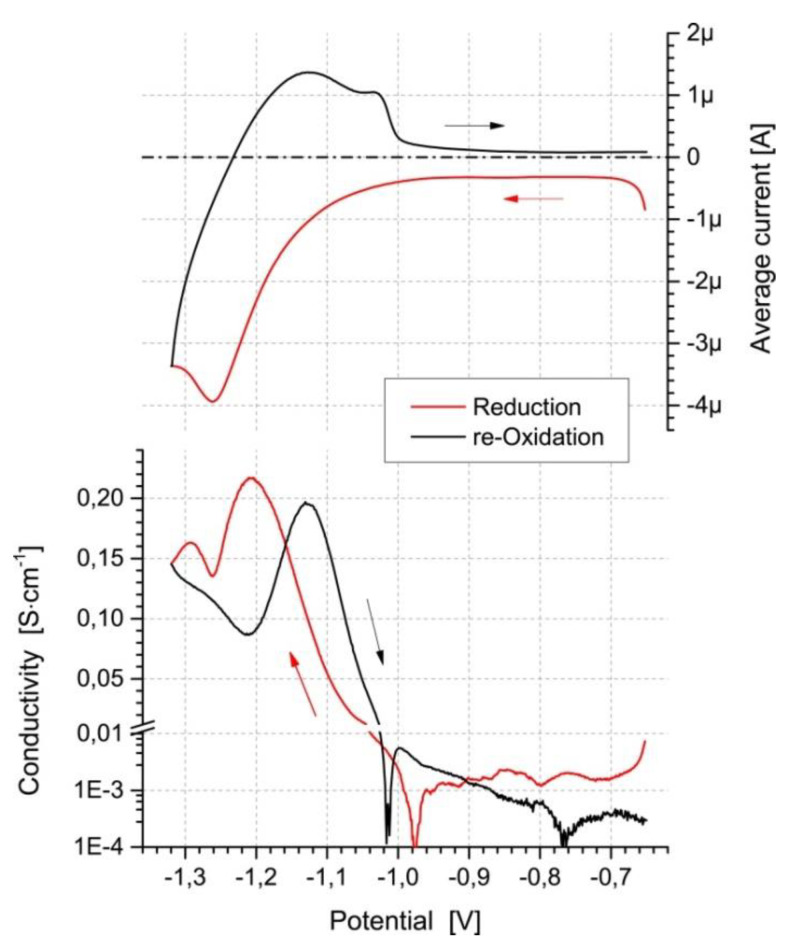
Time-resolved potential and compound (pseudo-CV) current and conductivity plots for PNDI-C deposited at interdigitated Pt/glass comb electrode pair. Potential scanning rate: 1 mV s^−1^, working electrode offset potential: 5 mV, 0.1 M Bu_4_NPF_6_/ACN supporting electrolyte. Reprinted from [65] under a Creative Commons license CC BY 4.0.

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
