# Peer review of "The Role of Electrochemical and Spectroelectrochemical Techniques in the Preparation and Characterization of Conjugated Polymers: From Polyaniline to Modern Organic Semiconductors"

_polymers, 2022, doi:10.3390/polym14194173_

Round 1

Reviewer 1 Report

This manuscript summarizes the electrochemical and spectroelectrochemical techniques used in the research process of conjugated polymers. The topic is novel and instructive, facilitating the development of a thorough understanding of conjugated polymers by advanced techniques and means. Therefore, I would like to recommend this manuscript to be published in this journal after addressing the following issues.

1. The section of ‘2. Electrochemical and spectroelectrochemical techniques used in the preparation and characterization of conjugated polymers’ can be elaborated with more references. For instance, concrete examples are needed to illustrate the application of the methods to make the text more comprehensive in ‘2.3. Spectroelectrochemical methods, 2.4. Time-resolved spectroelectrochemical methods.

2. The Conclusion is suggested to put forward your vision for future work, or the future research direction that needs to improve and strengthen.

Author Response

Thank you for your kind review. The manuscript has been improved according your comments.

  1. The section of ‘2. Electrochemical and spectroelectrochemical techniques used in the preparation and characterization of conjugated polymers’ can be elaborated with more references. For instance, concrete examples are needed to illustrate the application of the methods to make the text more comprehensive in ‘2.3. Spectroelectrochemical methods, 2.4. Time-resolved spectroelectrochemical methods.

Response: As suggested the number of references and examples has been increased. Please see updated subsections 2.3. Spectroelectrochemical methods and 2.4. Time-resolved spectroelectrochemical methods.

  1. The Conclusion is suggested to put forward your vision for future work, or the future research direction that needs to improve and strengthen.

Response: The Conclusion section has been expanded. A new paragraph in which we defined perspectives has been added

Reviewer 2 Report

Conjugated polymers are a class of important materials for modern organic semiconductors. To investigate these polymers, electrochemical and spectroelectrochemical techniques are critical. In this manuscript, the authors summarize these techniques for synthesizing and understanding conjugated polymers and some examples are given. It is a thorough and useful review which will be very helpful in the field of polymers. It is recommended for publication in Polymers. The followings are some minor questions.

1.      The full names of abbreviations should be given at their first appearance such as UV-Vis-NIR, OLEDs, ECE mechanism.

2.      Limitations of the electrochemical and spectroelectrochemical techniques could be given so that readers can have a complete understanding of them. For example, which type of conjugated polymers can be made by electrochemical techniques, which ones cannot.

3.      Conjugated polymers have been developed for 40 years. What’s next? Could the authors provide some perspectives on them and the synthesis and characterization techniques.

Author Response

Thank you for your kind review. The manuscript has been improved according your comments.

  1. The full names of abbreviations should be given at their first appearance such as UV-Vis-NIR, OLEDs, ECE mechanism.

Response: This has been completed, please check the text of the manuscript.

  1. Limitations of the electrochemical and spectroelectrochemical techniques could be given so that readers can have a complete understanding of them. For example, which type of conjugated polymers can be made by electrochemical techniques, which ones cannot.

Response: The discussion in subsections 2.1 Electrochemical polymerization and 2.3 Spectroelectrochemical methods have been extended. New references and examples has been added. Please see updated manuscript text.

  1. Conjugated polymers have been developed for 40 years. What’s next? Could the authors provide some perspectives on them and the synthesis and characterization techniques.

Response: The Conclusion section has been expanded. We referred to the expected increase in the role of electrochemical production of polymers and the development of polymers as modern semiconductors.